# Using the Dual Concept of Evolutionary Game and Reinforcement Learning in Support of Decision-Making Process of Community Regeneration—Case Study in Shanghai

**Youmei Zhou** [1] , **Hao Lei** [2] , **Xiyu Zhang** [1] , **Shan Wang** [3] , **Yingying Xu** [4] , **Chao Li** [5,*] and **Jie Zhang** [6,*]

1  Department of Landscape, College of Architecture and Urban Planning Tongji University, Shanghai 200092, China
2  Department of Landscape, The University of Sheffield, Sheffield S10 2TN, UK
3  Graphic Information Center, Shanghai University of Medicine & Health Sciences, Shanghai 201318, China
4  School of Humanities and Social Science, Beihang University, Beijing 100191, China
5  Zhijiang College, Zhejiang University of Technology, Shaoxing 312030, China
6  School of International Communications and Education, Communication University of Zhejiang, Hangzhou 310018, China
*  Correspondence: cynthia0217@163.com (C.L.); 19950482@cuz.edu.cn (J.Z.)

**Abstract:** Under the digital revolution that spawned in recent years, AI support is raised in the context of urban design and governance as it aims to match the operation of the urban developing process. It offers more chances for ensuring equality in public participation and empowerment, with the possibility of projection and computation of integrated social, cultural, and physical spaces. Therefore, this research explored how scenario simulation of social attributes and social interaction dimensions can be incorporated into digital twin city research and development, which is seen as a problem to be addressed in the refinement and planning of future digital platforms and management in terms of decision-making. To achieve the research aim, this paper examined the evolution of social governance state and strain decision models, built a simulation method for the evolution of complex systems of social governance driven by the fusion of data and knowledge, and proposed a system response to residents' ubiquitous perception and ubiquitous participation. The findings can help inspire the application of computational decision-making support in urban governance, and enhance the internal drive for comprehensive and sustainable urban regeneration. Moreover, they imply the role of the updated iterations of physical space and social interaction on social attributes.

**Keywords:** AI technology-driven community regeneration; decision-making; evolutionary game; reinforcement learning




## 1. Introduction

The development of computer science and information technology has been accompanied by fundamental changes in the industrial and production structure of the global economy, resulting in a markedly different social formation from the governance logic of industrial societies. As Alvin Toffler suggested in Cyberspace [1], stakeholders are given a stronger voice and governance structure. A new governance model will gradually emerge in which multiple actors such as government, market, and society cooperate equally and share power [2]. Digital technology can effectively improve the efficiency of multi-sectoral and multi-subject cooperation and provide technical support for collecting, processing, and applying the information to assist decision-making.

However, due to the lack of proper designation of digital methods for efficient governance, there is a need for increased research on AI technology-driven community regeneration, especially in the new technological context and governance paradigm. Under such a context, the research aimed to provide key technologies for digital governance at the social level, enabling object-based development and scenario deepening, and providing

technical and data support for the comprehensive construction of a smart city with deep learning and self-optimization capabilities. Based on computational social science and evolutionary game theory, the study examined the evolution of demand and responsibility decisions in community regeneration, constructed an evolutionary simulation approach for complex systems of social governance driven by the fusion of data and knowledge, and extrapolated, validated, and optimized the abovementioned system framework and the overall architecture of a digital twin system based on big data for social governance. Therefore, this study aimed to improve the workflow of urban planning and management decision-makers, provide a more effective way for the public to participate in community construction and management, promote community resource sharing and environmental co-creation, and increase residents' attention to and participation in community public affairs, thereby realizing a smart transformation of community renewal.

## 2. Literature Review

### 2.1. The Impact of Digital Technology on the Improvement of Urban Management Refinement

Digital technologies and digital twins are driving dramatic changes in urban governance. In recent years, more and more countries are applying digital twin technology to the construction of smart city decision-making platforms [3]. For example, Singapore has built a city operation simulation system called CityScope, which enables the simulation and optimization of cities and planning decisions. The CSIC Research Centre at the University of Cambridge, the CASA Centre at the University of London, and the MIT sensible city laboratory in the US have also made corresponding explorations. However, digital twin-based simulations are still focused on the physical space level, and no effective theoretical support and practical exploration have been formed in the social perception dimension [4]. Digital twin was first introduced in the early 21st century by Michael Grieves [5], whose expertise in product design initially rooted the concept in production engineering. In city level, the digital twin acts as a digital simulation technology that has been increasingly used in the field of urban governance to provide multi-scenario simulation and decision support for cities [3]. The mirror image of a physical process usually matches exactly the operation of the physical process that occurs in real-time [6]. In this context, the idea of a digital twin emerged from representing the physical asset side of the city. Geographic information systems, scaled down to the level of buildings and extended by building information modeling software to deal with the operation of buildings in terms of energy, material use, and maintenance, provide the context for a broad digital representation at the level of all physical assets in the city [7]. The digital twin refers to the data analysis and modelling of physical entities to form a multi-disciplinary, multi-physical, multi-time scale and multi-probability simulation process [8], which can reflect the whole life cycle process of physical systems in different realistic scenarios and is a key technology to realize the mapping of physical systems to digital models in information space. In the field of Community Regeneration, the simulation of multi-stakeholder demands inevitably involves the examination of the social dimension [9]. Therefore, the quantitative input and simulation of the full range of data are necessary for the digital twin service community regeneration.

In the digital age, residents' lives and behavior patterns have changed dramatically, and their diverse needs for improved community environments, public services, and facilities have increased [10]. The traditional community governance process is a linear top–down model characterized by the linkage of "planning and design decision-making implementation feedback and optimization". With the problems of a single decision-maker and relatively late information, the traditional community environment and its services cannot effectively meet the needs in terms of supply of community public services, real-time collection, and processing of residents' needs [11]. Moreover, redeveloping the contemporary community requests dynamic tracking and feedback on the decision-making process and implementation status of Community Regeneration, and it is necessary to form real-time identification of the operation status of the community management system and preventive monitoring of potential problems [12]. As a result, the research and practice of

Community Regeneration are lagging behind, and a new research paradigm needs to be developed based on digital technology. Therefore, how to dovetail the overall conceptual framework of digital twin city and find reasonable methods and decision paths for decision support in specific scenarios that take into account complex sociological issues, such as community regeneration, becomes the main question.

*2.2. Decision-Making Methods and Community Regeneration*

In the field of community regeneration, effective and speedy quantitative models for multi-party demand analysis and response have become a direction worth exploring, and evolutionary game theory and deep learning are of great importance to this problem based on the simulation of full-volume data for assisted decision-making [13]. Community regeneration is a fundamental area of urban governance and is generally based on balancing interests and maximizing public benefits to improve the built environment of a community. In this regard, it requires decision makers to weigh the needs of various stakeholders and to review and synthesize information from physical spaces, interactive behaviors, and social dimensions.

Theoretical approaches relating to AI technologies, such as game theory and digital twins, are becoming popular in urban governance, but there is still a lack of systematic and implementable solutions. The positive effects of AI in public service provision and administrative efficiency have been well documented and theoretical models have been constructed, which can be divided into three main categories: information economics, game theory based on rational behavior, and behavioral economics based on irrational behavior. On the other hand, at the level of mechanism design, Leonid Hurwicz, winner of the 2007 Nobel Prize in Economics, has also proposed a mechanism design theory. However, behavioral economics and other decision models are primarily used as mathematical models in transport and energy, and there are high barriers to the interface and application to real-life scenarios close to the population. In the field of community regeneration, the lack of in-depth theoretical models for multi-group interest games.

Evolutionary game theory is a further development of game theory, and it predicts the strategy choice of each subject. It also simulates the game process between subjects to develop the optimal strategy to maximize the group's interests and achieve the Nash equilibrium, and finally form the corresponding auxiliary decision-making recommendations [14]. Evolutionary game theory is widely used in the study of social institutions, behavioral norms, and other socio-economic issues due to the fact that the dynamic choice mechanism it uses can be applied to multiple games of the system and form an evolutionary and stable strategy [10,15]. In the field of community regeneration, some studies have analyzed the best way for government departments and social capital to participate based on a game perspective, which also shows that evolutionary game theory is equally applicable to community stakeholders because there is a continuous game process [8,16]. The evolutionary game approach explores the game mechanism between government departments' regulatory strategy and developers' service quality [17]. In summary, scholars have used evolutionary game theory to study the interest claims and interactions of interest subjects and construct a collaborative cooperation mechanism, providing a theoretical basis and practical guidance for this study.

Traditional operations optimization methods are the main means to solve combinatorial optimization problems, but with the increasing size of problems and the demand for real-time solutions, traditional OOP algorithms are under great computational pressure, and it is difficult to solve combinatorial optimization problems online [18]. In recent years, with the rapid development of deep learning technology, deep reinforcement learning has attracted attention in the fields of Go and robotics. The results have shown its powerful learning ability and sequential decision-making capability [19]. Deep reinforcement learning is a new approach to solving combinatorial optimization problems, with fast solution speed and strong model generalization advantages [20]. Furthermore, it provides a new way of aiding decision-making and tools for decision optimization in the intersection of

social complexity science problems, such as urban governance and urban regeneration, and how to make the best cooperative strategies under different conditions in community regeneration [3,21].

*2.3. The Conceptual Framework of the Decision-Making Process of Community Regeneration Proposed by Evolutionary Game and Reinforcement Learning*

The project builds a decision support framework system that combines reinforcement learning and evolutionary games based on the game structure of stakeholders developed by multiple researchers [22,23]. The basic response mechanism and influence mechanism of smart community renewal are built based on the collection and analysis of scenarios, cases, strategies, and utilities from big data, and the game behavior is identified [4]. The behavioral mechanisms and responses of complex decision trees can be optimized for urban regeneration and community regeneration where information is incomplete, and training cases are limited. Ultimately, the monitoring of multi-dimensional holographic data in specific scenarios and the prediction of scenario simulations, through the virtualization of data models intervene in the digital twin community simulation platform to realize a system that provides optimal decision support based on environmental conditions and feedback autonomously, providing an optimal phased strategy for the government to reduce economic burdens, improve the well-being of residents and promote the long-term internal drive of enterprises to participate in community regeneration [24,25]. Based on this combination of analysis and strengths, we designed and proposed a conceptual framework for a reinforcement learning solution as shown in the figure.

As shown in Figure 1, we used gaming as the Agent for adaptive reinforcement learning, forming the environment in the model through perceptual data about the environment, and forming the action on the elements of the environment through the learning of categorical case data. This action is also a translation of the optimal decision from the agent's output, which will be a feedback based on the current concept of benefits. Although there is still a lack of documented data on this, for the future, residents will focus more on public participation and self-fulfilling experiences, while the large amount of data and experience on how to respond and model people's participation will be the focus of future decisions. The model is an AI-assisted decision-making framework, which is based on a multidisciplinary integrated view of the changes in human perceptions and emotional ties in the renewal process. Moreover, it is based on a sociological classification of residential–government–business partnerships and integrated built environment research. This research aimed to architect a framework of learning mechanisms and scenarios for smart community renewal and has an important role in the development and evolution of future data infrastructures. The decision-making requirements will guide the data storage and collection. The light green part inside the agent will be the core of this system model and this thesis will test and fit the feasibility of this part of the evolutionary game algorithm for simulated predictions in community renewal and in doing so demonstrate the applicability of the method as a core algorithm.

Based on this combination of analysis and strengths, we designed and presented a conceptual diagram of the way in which reinforcement learning and evolutionary games can be combined. In Figure 2, we describe the basic response and influence mechanisms, and to identify the game behaviors involved. Then, we built a database of strategies and evolutionary games, abstracted a scenario-based evolutionary game model, and simulated complex decision trees by using machine learning neural networks algorithms. There are still complex sociological issues, such as the different driving or leading models in different communities, which affect the design of their agent's perceptual model and parameters and indicators [21,26]. This research has an important role in forming a learning mechanism and scenario framework for updating the smart community in constructing future data infrastructures. Decision-making will then help guide data storage and collection.

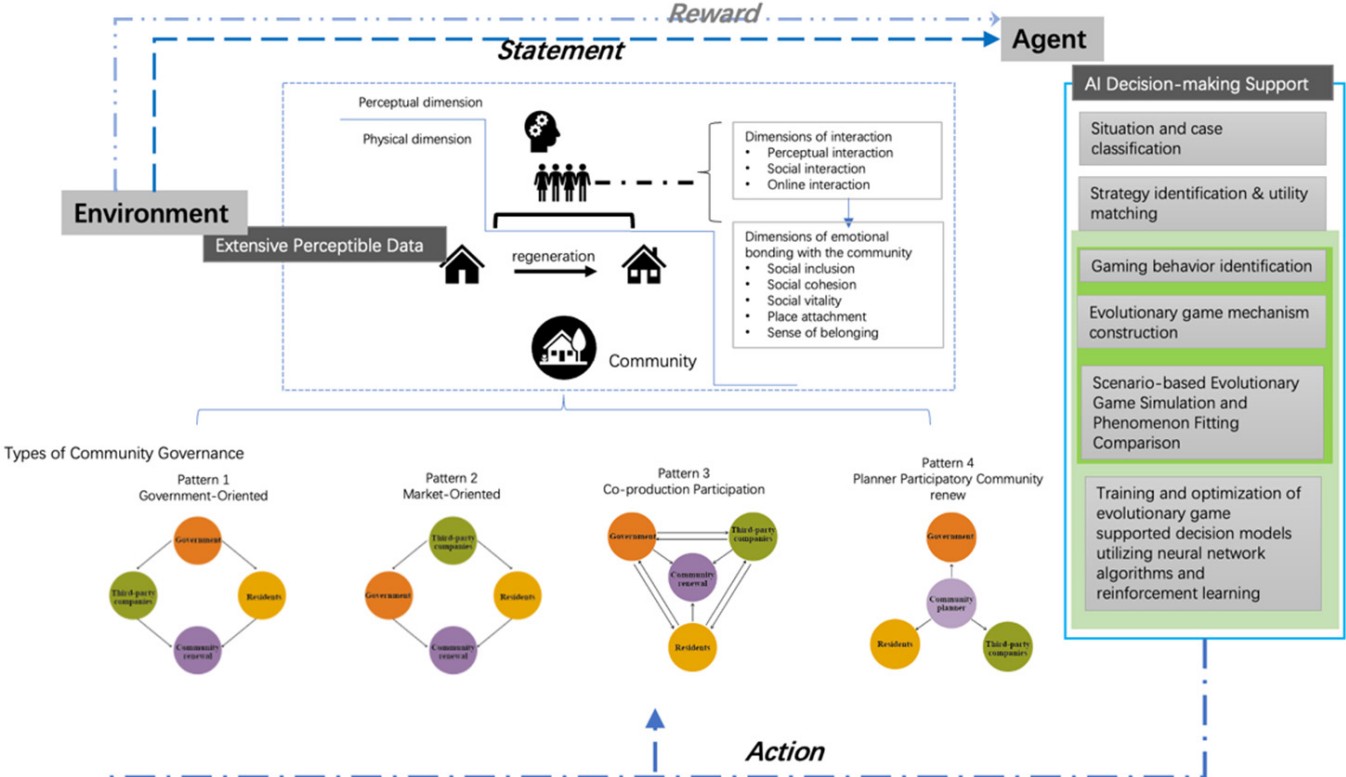

**Figure 1.** The conceptual framework of the decision-making process of community regeneration proposed by evolutionary game and reinforcement learning.

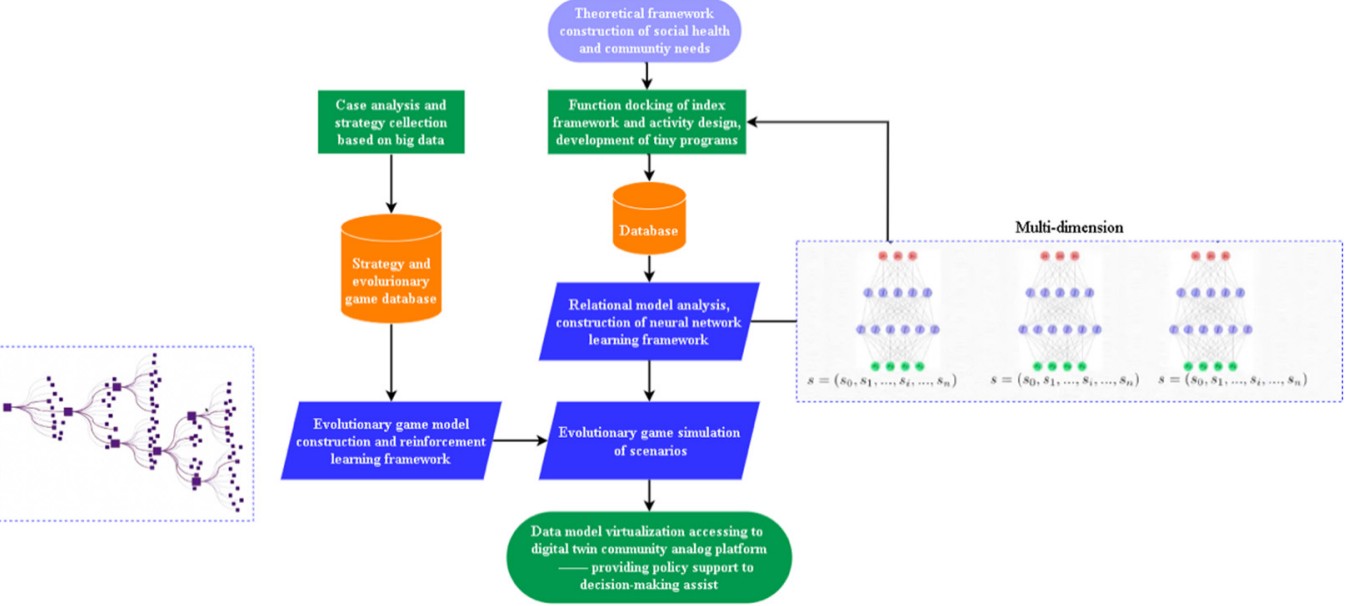

**Figure 2.** The conceptual framework of using machine learning neural networks algorithms and reinforcement learning.

We used game decision-making as the agent of reinforcement learning. The projection of ubiquitous perceptual data in the digital twin forms the environment in the model. Learning case data forms the action for the environmental elements, which will be a beneficial concept based on the present. However, due to the limitation of the previous

technology, there are not enough data of social dimension relating to the digital twin city and community in the cyber age currently [24,26], which means that we should plan to do data burial in future data collection based on the framework concept of the simulation. As Jana et al. (2020) suggested, residents are more focused on public participation and self-experience, and how to process and simulate a large amount of data and experiences of the involvement of people will be the focus of future decisions [27]. The light green part of the agent in Figure 1 will be the core part of this system model, and this thesis will test and fit the feasibility of the core part of the evolutionary game algorithm for simulation prediction in community renewal, in order to demonstrate the applicability of this method as the core algorithm. This study is based on the framework of one of these scenarios and forms the first part of the project.

## 3. Method

Based on the characteristics of the different community types sorted out above, this paper selected the innovative community planner system type (type 4 in Figure 1), used age-friendly renewal as a scenario for evolutionary game simulation, and carried out modeling and simulation to project back governance strategies through analysis of stakeholder benefit relationships in order to verify the system model proposed in this program in complex social problems.

### 3.1. Model-Related Hypotheses, Parameters, and Payoffs

The community planner system aims to achieve a balance of multiple interests in the practice of age-friendly renewal. To optimize the realization path of this goal and promote the high-quality development of aging-friendly regeneration, an evolutionary game framework was introduced to build a tripartite evolutionary game model consisting of government, enterprises, and residents to maximize the private interests of each subject and the interests of the multi-subjects' shared governance network. Among them, government subjects mainly include administrative departments related to residential area renovation and neighborhood committees; enterprises mainly include third-party social organizations such as property and infrastructure developers; residents mainly refer to residents of all ages in old communities (including tenants and indigenous people).

Based on the analysis of game subjects, this study makes the following hypotheses.

- Hypothesis 1: The government, enterprises, and residents are all finite rational, and will repeatedly adjust their strategies according to their benefits, and finally choose the optimal strategy to maximize their interests.
- Hypothesis 2: The government's strategy choice is (positive governance, loose governance). Positive governance refers to the government's full interference in the promotion of work, including subsidies, publicity, work promotion, etc. Additionally, through rewards and punishments, the government promotes social organizations and residents to actively participate in community renewal. Loose governance means that the government does not interfere too much in the process of work.
- Hypothesis 3: The strategy choice of enterprises is (active participation, negative participation). Active participation means that social organizations provide quality services to residents and can enjoy government subsidies; negative participation means that social organizations maintain the status quo and have low participation.
- Hypothesis 4: The strategy choice of residents is (active participation, negative participation). Active participation means that residents are in favor of community renewal and participate in it to improve their quality of life, to express their ideas and join in the activities, while negative participation means that residents are indifferent to the process, and the benefits they obtain depend only on the strategic choices of the government and enterprises.

The benefits and costs differ when each entity chooses a different strategy. When the government chooses positive governance, it needs to give subsidies to enterprises and residents (S). If enterprises adopt negative strategies at this time, they will impose certain

penalties on market subjects (M). In the process of age-appropriate renewal, no matter what kind of policy strategy is adopted, it is necessary to pay the cost of the plan preparation in the early stage, the promotion cost in the process of the plan implementation, and the plan evaluation after the plan implementation ($C_e$). When the market players cooperate with the government's strategy implementation, the government–enterprise cooperation will save a certain percentage ($\beta$) of the cost for the government. The government department will gain direct and indirect benefits such as economical as well as prestige when community governance makes some progress ($P_1$). However, in the case of loose governance, the poorer living experience of residents leads to a certain loss of prestige for the government ($P_2$, $P_1 > P_2$). When government actively governances and residents are actively involved, community planners will be recruited to participate in community renewal in order to optimize the original services, which will reduce the cost of community renewal and further improve the infrastructure and quality of life of residents. At this time, the cost paid by the government will be reduced by $\alpha$ times.

If market players choose the active participation strategy, the cost they need to pay ($C_1$) is much larger than the cost they need to pay when they adopt the negative participation strategy ($C_2$), i.e., $C_1 > C_2$. When community governance achieves certain progress, market players such as developers will gain additional economic benefits due to the increase of reputation in the industry in addition to the basic benefits ($R_f$) from providing public services ($R_e$). While market agents adopt a negative participation strategy, they need to pay additional fees to compensate residents for the damage caused to their interests ($L_p$).

When a resident subject chooses an active participation strategy, they will receive not only government subsidies, but also indirect benefits in terms of physical, mental, and life quality improvement ($R_i$), but they need to pay for the services provided by the market subject ($R_f$). However, when the implementation of the community renewal program is not thorough, the quality of life of residents may be reduced, resulting in the loss of residents' interests ($L_n$). When the government actively supervises, and residents actively participate and provide feedback on the infrastructure user experience and demand changes, it is convenient for community planners to optimize existing facilities and give residents a better life experience. At this time, residents' income will increase by $\alpha$ times.

From the abovementioned assumptions, the parameters in the three-party game process were set as shown in the following table.

### 3.2. Model Framework and Solution

In this paper, we assumed that the probabilities of government, market, and resident subjects choosing positive strategies are $x_i, y_j, z_k$ ($x_i > 0, y_j > 0, z_k > 0$), where $i \in S_G$, $j \in S_M, k \in S_R$, and $\sum_{i \in S_G} x_i = 1$, $\sum_{j \in S_M} y_j = 1$, $\sum_{k \in S_R} z_k = 1$.

In the game process, each subject is finite rational, i.e., it will change its strategy choice according to the comparison with the gain of other subjects, so the game process has a period, and this paper took $N(t)$ as the total number of sectors (institutions or residents) of subjects, and $N_i(t)$ as the number of sectors (institutions or residents) that choose active strategies at time $t$. Taking the government subject as an example, then $x_i(t) = N_i(t)/N(t)$ represents the probability that the subject picks an active strategy at moment $t$. Therefore, the government gain is $(x, x) = \sum_{i \in S_G} x_i(t) \cdot u(i, x)$. Based on Table 1, the three-party game payoff matrix can be derived as follows.

**Table 1.** Parameter setting table of the three-party game.

| Parameter | Meaning | Remarks |
|---|---|---|
| $S$ | Subsidies given to businesses and residents when the government is aggressive in governance | |
| $C_e$ | Costs paid by the government when it adopts a proactive strategy | |
| $M$ | Penalties for negative participation of enterprises when the government is active in governance | |
| $\alpha$ | The percentage of cost saved by the government and residents' income increased when community planners are involved in the project | $\alpha \in [0,1]$ |
| $\beta$ | The proportion of cost savings for the government and enterprises when the government and enterprises cooperate | $\beta \in [0,1]$ |
| $P_1$ | Direct and indirect benefits such as economic and prestige gained by government departments when community governance is progressed | |
| $P_2$ | Loss of prestige due to poor living experience of residents when the government is passive | $P_1 > P_2$ |
| $C_1$ | Costs to businesses when they actively participate | |
| $C_2$ | The cost of active participation by the government and passive participation by enterprises | $C_1 > C_2$ |
| $R_f$ | The basic benefits of public services provided by market players | |
| $R_e$ | Additional economic benefits for each market player due to increased reputation in the industry | |
| $R_i$ | Indirect benefits to residents in terms of physical, mental, and quality of life improvement | |
| $L_p$ | Compensation costs for residents who suffer as a result of the negative participation strategy of market players | |
| $L_n$ | The benefit loss of residents when the community renewal effect is poor | |

Based on the assumptions and Table 2, we can know the equation of benefits when the government, companies, and residents choose positive and negative strategies respectively.

$$
\begin{aligned}
u(i,x) = &\ y_j z_k (P_1 - S - (1-\alpha-\beta)C_e) + (1-y_j)z_k(P_1 - S + M - (1-\alpha)C_e) \\
&+ y_j(1-z_k)(P_1 - S - (1-\beta)C_e) + (1-y_j)(1-z_k)(M - S - C_e)
\end{aligned} \tag{1}
$$

$$
\begin{aligned}
u(i,1-x) = &\ y_j z_k(-C_e - P_2) + (1-y_j)z_k(-C_e - P_2) + y_j(1-z_k)(-C_e)' \\
&+ (1-y_j)(1-z_k)(-C_e)
\end{aligned} \tag{2}
$$

$$
\begin{aligned}
u(j,y) = &\ x_i z_k\left(S + R_f + R_e - C_1\right) + x_i(1-z_k)(S + R_e - C_1) + (1-x_i)z_k\left(R_f + R_e - C_1\right)^4 \\
&+ (1-x_i)(1-z_k)(-C_1)
\end{aligned} \tag{3}
$$

$$
\begin{aligned}
u(j,1-y) = &\ x_i z_k\left(S - M + R_f - C_2 - L_p\right) + x_i(1-z_k)(S - M - C_2)' \\
&+ (1-x_i)z_k\left(R_f - L_p - C_2\right) + (1-x_i)(1-z_k)(-C_2)
\end{aligned} \tag{4}
$$

$$
\begin{aligned}
u(k,z) = &\ x_i y_j\left(\alpha R_i - R_f\right) + x_i(1-y_j)\left(\alpha R_i + L_p - R_f - L_n\right) + (1-x_i)y_j\left(R_i - R_f\right) \\
&+ (1-x_i)(1-y_j)\left(-L_n - R_f\right)
\end{aligned} \tag{5}
$$

$$
u(k,1-z) = x_i y_j(R_i) + x_i(1-y_j)(-L_n) + (1-x_i)y_j(-L_n) + (1-x_i)(1-y_j)(-L_n) \tag{6}
$$

**Table 2.** The matrix of benefits of the three-party game.

| Strategy Selection | | | Earnings | | |
|---|---|---|---|---|---|
| Government | Companies | Residents | Government | Companies | Residents |
| C | C | C | $P_1 - S - (1-\alpha-\beta)C_e$ | $S + R_f + R_e - C_1$ | $\alpha R_i - R_f$ |
| C | C | D | $P_1 - S - (1-\beta)C_e$ | $S + R_e - C_1$ | $R_i$ |
| C | D | C | $P_1 - S + M - (1-\alpha)C_e$ | $S - M + R_f - C_2 - L_p$ | $\alpha R_i + L_p - R_f - L_n$ |
| C | D | D | $M - S - C_e$ | $S - M - C_2$ | $-L_n$ |
| D | C | C | $-C_e - P_2$ | $R_f + R_e - C_1$ | $R_i - R_f$ |
| D | C | D | $-C_e$ | $-C_1$ | $-L_n$ |
| D | D | C | $-C_e - P_2$ | $R_f - L_p - C_2$ | $-L_n - R_f$ |
| D | D | D | $-C_e$ | $-C_2$ | $-L_n$ |

$u(i, x), u(j, y), u(k, z)$ represents the expected returns under the positive strategy chosen by the government, companies, and residents, respectively; $u(i, 1 - x), u(j, 1 - y), u(k, 1 - z)$ represent the expected returns under the choice of negative strategies by the three main game players, respectively.

Define $u(i, x) - u(i, 1 - x)$ as the matrix A:

$$A = \begin{pmatrix} P_1 - S + (\alpha + \beta)C_e + P_2 & P_1 - S + \beta C_e \\ P_1 - S + M + \alpha C_e + P_2 & M - S \end{pmatrix} \tag{7}$$

Similarly, the matrices B and C can be obtained as:

$$B = \begin{pmatrix} R_e - C_1 + M + L_p + C_2 & R_e - C_1 + M + C_2 \\ R_e - C_1 + L_p + C_2 & C_2 - C_1 \end{pmatrix} \tag{8}$$

$$C = \begin{pmatrix} (\alpha - 1)R_i - R_f & \alpha R_i + L_p - R_f \\ R_i - R_f + L_n & -R_f \end{pmatrix} \tag{9}$$

According to the theory of replicative dynamics during the evolutionary game, the group of game subjects who choose strategy I will adjust their strategy selection to maximize their payoffs based on the difference between their payoffs and the average payoffs, as shown in the following equation.

$$\frac{dx_i}{dt} = x_i \cdot [u(i, x) - u(x, x)] \tag{10}$$

The dynamic equation of the three-party game is known from Equation (7) as follows.

$$\frac{dx_i}{dt} = x_i(1 - x_i)\left(y^T A z\right) = x_i(1 - x_i)\left(\sum_{j \in S_D} \sum_{k \in S_R} y_j a_{jk} z_k\right) \tag{11}$$

$$\frac{dy_j}{dt} = y_j(1 - y_j)\left(z^T B x\right) = y_j(1 - y_j)\left(\sum_{k \in S_R} \sum_{i \in S_G} z_k b_{ki} x_i\right) \tag{12}$$

$$\frac{dz_k}{dt} = z_k(1 - z_k)\left(x^T C y\right) = z_k(1 - z_k)\left(\sum_{i \in S_G} \sum_{j \in S_D} x_i c_{ij} y_j\right) \tag{13}$$

The replicated dynamic equations for the three-party game in community renewal were obtained by bringing the above-analyzed matrix and parameters into the abovementioned equations and making $F_1(x_i, y_j, z_k) = dx_i/dt$, $F_2(x_i, y_j, z_k) = dy_j/dt$, $F_3(x_i, y_j, z_k) = dz_k/dt$ are shown below.

$$\begin{aligned} F_1(x_i, y_j, z_k) = x_i(1 - x_i)\big(y_j z_k(P_1 - S + (\alpha + \beta)C_e + P_2) + (1 - y_j)z_k + (P_1 - S + M + \alpha C_e + P_2) \\ + y_j(1 - z_k)(P_1 - S + \beta C_e) + (1 - y_j)(1 - z_k)(M - S)\big) \end{aligned} \tag{14}$$

$$\begin{aligned} F_2(x_i, y_j, z_k) = y_j(1 - y_j)\big(x_i z_k(R_e - C_1 + M + L_p + C_2) + x_i(1 - z_k)(R_e - C_1 + M + C_2) \\ + (1 - x_i)z_k(R_e - C_1 + L_p + C_2) + (1 - x_i)(1 - z_k)(C_2 - C_1)\big) \end{aligned} \tag{15}$$

$$\begin{aligned} F_3(x_i, y_j, z_k) = z_k(1 - z_k)\Big(x_i y_j\big((\alpha - 1)R_i - R_f\big) + x_i(1 - y_j)\big(\alpha R_i + L_p - R_f\big) \\ + (1 - x_i)y_j\big(R_i - R_f + L_n\big) + (1 - x_i)(1 - y_j)\big(-R_f\big)\Big) \end{aligned} \tag{16}$$

Subsequently, according to the nature of replicated dynamic equations, the evolutionary equilibrium points of the game model were found by making $F_1(x_i, y_i, z_i) = 0$, $F_2(x_i, y_i, z_i) = 0$, $F_3(x_i, y_i, z_i) = 0$, respectively.

$$\begin{cases} F_1(x_i, y_j, z_k) = x_i(1 - x_i)\left[ \begin{array}{c} (P_1 - M - \beta C_e - P_1 z)y \\ +(P_1 + P_2 + \alpha C_e)z + M - S \end{array} \right] = 0 \\ F_2(x_i, y_j, z_k) = y_j(1 - y_j)\left[ \begin{array}{c} (M + R_e - (M - L_p + R_e)x)z \\ +(M + R_e)x + C_2 - C_1 \end{array} \right] = 0 \\ F_3(x_i, y_j, z_k) = z_k(1 - z_k)\left[ \begin{array}{c} (L_n + \alpha R_i - (L_n + L_p + 2R_i)y)x \\ +(L_n + R_i)y - R_f \end{array} \right] = 0 \end{cases} \quad (17)$$

Taking $F_2(x_i, y_j, z_k) = 0$ as an example, when $Z_k \neq \frac{(M+R_e)x+C_2-C_1}{(M-L_p+R_e)x-R_e-M}$, $y_j = 0$ or $y_j = 1$. And $\frac{dF_2(x_i, y_j, z_k)}{dy_j} = (1 - 2y_j)\left[(M + R_e - (M - L_p + R_e)x)z + (M + R_e)x + C_2 - C_1\right]$. When $Z_k > \frac{(M+R_e)x+C_2-C_1}{(M-L_p+R_e)x-R_e-M}$, $F_2'(x_i, 0, z_k) > 0$, $F_2'(x_i, 1, z_k) < 0$, 1 is a stable solution, i.e., when the probability of the government actively regulating community renewal $Z_k > \frac{(M+R_e)x+C_2-C_1}{(M-L_p+R_e)x-R_e-M}$, the probability of the firm adopting an active strategy will increase. Finally, the government and enterprises will cooperate with each other. The probability of strategy increases and both parties eventually reach cooperation; when $Z_k < \frac{(M+R_e)x+C_2-C_1}{(M-L_p+R_e)x-R_e-M}$, $F_2'(x_i, 0, z_k) < 0$, $F_2'(x_i, 1, z_k) > 0$, 0 is the stable solution, i.e., when the probability of government actively regulating community renewal $Z_k < \frac{(M+R_e)x+C_2-C_1}{(M-L_p+R_e)x-R_e-M}$, the probability of firms adopting an active strategy decreases until it is 0. When $Z_k = \frac{(M+R_e)x+C_2-C_1}{(M-L_p+R_e)x-R_e-M}$, $F_2(x_i, y_j, z_k) = 0$. At this time, the government and residents will choose either pure or mixed strategies, thus forming a mixed-strategy equilibrium point for the three subjects. The equilibrium points of the three-way game system can be solved as shown in Appendix A.

Due to $x, y, z \in [0, 1]$, so $E_9 \sim E_{13}$ make sense under certain conditions. Because $S - P_1 - \beta C_e < 0$, and $C_1 - C_2 - M - R_e < 0$, so $E_{10}$, $E_{13}$ make no sense.

According to Lyapunov's first theorem, it can be known that all the eigenvalues of Jacobian matrix (A2 in Appendix) have negative real parts, so the equilibrium point is asymptotic stable point. If at least one eigenvalue of Jacobian matrix has a positive real part, the equilibrium point is unstable. The Jacobian matrix has negative real parts except eigenvalues with zero real parts, so the equilibrium point is in critical state and the stability cannot be determined by the sign of eigenvalues. The stability analysis of each equilibrium point is shown in the following table.

As can be seen from Table 3, when $M$ is less than $S$ that is when the government's punishment to enterprises is less than the government's subsidy to enterprises, it is the gradual stability point; otherwise, it is the unstable point. It can be seen that when the punishment to enterprises is less than the subsidy, it is more likely to choose negative strategies. Therefore, a reasonable balance of reward and punishment measures is needed. When $S - P_1 + \beta C_e < 0$ that is when the government's social prestige and the rewards given by the superior government far outweighed the cost, $E_5$ is the stability point. To stabilize the situation, the government preferred complete management rather than loose management, and promoted the active participation of enterprises and residents through various measures to promote the community renewal work quickly. When $P_1 + P_2 - S + \alpha C_e - \beta C_e < 0$, $E_7$ is stability point. That is, when the government put the improvement of people's quality of life at first, they will invest lots of money (greater than their awards issued by superior government departments as well as social benefits), so as to promote the enterprises to actively participate in the transformation for the community residents to create happiness life. At this time, the tripartite game will reach equilibrium system. When $L_P - L_n + R_f + (1 - \alpha)R_i$, that is, when the decline of residents' quality of

life is very serious, the government will take measures of active governance and improve community infrastructure through some relatively tough measures and policies to achieve the goal of improving residents' quality of life. According to the simplification calculation, when $a_1 > 0$, $E_9$ is the unstable point. Due to that $2M + P_1 + P_2 + C_e - 2S > 0$, when $L_P - L_n > 0$, that is, when the compensation from enterprises to residents is greater than the loss of benefits such as the quality of life, $E_{11}$ is the stable point.

**Table 3.** The parameters matrix of benefits of the three-party game.

| Equilibrium Point | Eigenvalues of the Jacobian Matrix | | | Symbol of the Real Part | Stability Conclusion |
|---|---|---|---|---|---|
| | $\lambda_1$ | $\lambda_2$ | $\lambda_3$ | | |
| $E_1(0,0,0)$ | $C_2 - C_1$ | $M - S$ | $-R_f$ | $(-,-,-)$ | ESS |
| $E_2(1,0,0)$ | $S - M$ | $L_n - R_f + \alpha R_i$ | $C_2 - C_1 + M + R_e$ | $(X,0,+)$ | Instability point |
| $E_3(0,1,0)$ | $C_1 - C_2$ | $P_1 - S - \beta C_e$ | $L_n - R_f + R_i$ | $(+,-,+)$ | Instability point |
| $E_4(0,0,1)$ | $R_f$ | $C_2 - C_1 + M + R_e$ | $M + P_1 + P_2 - S + \alpha C_e$ | $(+,+,+)$ | Instability point |
| $E_5(1,1,0)$ | $S - P_1 + \beta C_e$ | $C_1 - C_2 - M - R_e$ | $L_n - L_P - R_f - (1-\alpha)R_i$ | $(-,-,-)$ | ESS |
| $E_6(1,0,1)$ | $R_f - L_n - \alpha R_i$ | $C_2 - C_1 + L_P + M + R_e$ | $S - P_1 - P_2 - M - \alpha C_e$ | $(-,+,-)$ | Instability point |
| $E_7(0,1,1)$ | $R_f - L_n - R_i$ | $C_1 - C_2 - M - R_e$ | $P_1 + P_2 - S + \alpha C_e - \beta C_e$ | $(-,-,+)$ | Instability point |
| $E_8(1,1,1)$ | $L_P - L_n + R_f + (1-\alpha)R_i$ | $S - P_1 - P_2 - \alpha C_e + \beta C_e$ | $C_1 - C_2 - L_P - M - R_e$ | $(-,-,-)$ | Instability point |
| $E_9(0,y_1,z_1)$ | $0$ | $0$ | $a_1$ | $(0,0,+)$ | Instability point |
| $E_{11}(x_1,0,z_2)$ | $0$ | $a_2$ | $a_3$ | $(0,-,-)$ | Uncertain point |
| $E_{12}(x_2,y_2,0)$ | $a_4$ | $a_5$ | $a_6$ | $(X,X,X)$ | Uncertain point |

For the details of $a_1 a_2 a_3$, see Appendix A.

According to the stability analysis of equilibrium point, the following inference can be drawn. When $M < S$, $S - P_1 + \beta C_e < 0$, and $L_p - L_n > 0$, $E_1$, $E_5$, $E_{11}$ are stable points. Although all three parties may adopt negative strategies, none of them can find their own role positioning at the early stage of the work, which leads to difficulties in advancing the work. As long as we learn from excellent examples and hold discussion meetings, this situation will be gradually changed. Subsequently, the government basically adopt positive governance strategies and play a leading role in promoting the work, but the participation of residents is low. Therefore, it is necessary to increase the participation of the two parties and jointly promote the community renewal by rewarding residents who actively participate and appropriately punishing enterprises who passively participate.

## 4. Evolutionary Game Analysis

The previous section formed an evolutionary game model for age-appropriate renewal based on relevant assumptions. To verify the realistic interpretation of the model, this section assigns and simulates the operation in the context of community age-appropriate renewal practice in Baoshan District, Shanghai. The district has a high degree of aging, a high population density, and has advanced the age-appropriate renewal, which has the reference of the whole process.

We take this area as an example to simulate the tripartite game process under the community planner system. By conducting the research in the area, this research aimed to study the game process, strategy selection, and game results. Moreover, we discovered the problems of aging-appropriate renewal through the model, and compared them with the actual renewal process in this area to improve the model. Finally, we determined the method's usefulness for real problems, and analyze and draw relevant policy recommendations.

### 4.1. Model Parameter Substitution

At the level of the gaming process, at the beginning of the renovation, the community took the lead in submitting a renovation plan to the district government and received approval for the plan and special funding. The program requested that the water supply enterprises bear the material costs in the renovation process, the community's special housing maintenance fund bear the cost of commercial renovation, and the shortfall should be borne by government finance. At the economic level, this leads to a lack of one-time funding for the construction aspect of age-appropriate renewal. However, the construction companies themselves have difficulties in capital liquidity and are unable and unwilling to advance funds for construction. It led to a conflict between the government, the companies, and the residents. The community organized the game in the form of multi-party talks. Moreover, it took three stakeholder meetings to finalize the financing and distribution of the construction funds, the initial values of parameters in this study designed as Table 4.

**Table 4.** Initial values of the parameters of the three-party game.

| Parameter | Initial Value | Remarks |
|:---:|:---:|:---:|
| $S$ | 20 | |
| $C_e$ | 30 | |
| $M$ | 15 | |
| $\beta$ | 0.2 | $\beta \in [0, 1]$ |
| $\alpha$ | 0.15 | $\alpha \in [0, 1]$ |
| $P_1$ | 55 | |
| $P_2$ | 20 | $P_1 > P_2$ |
| $C_1$ | 30 | |
| $C_2$ | 10 | $C_1 > C_2$ |
| $R_f$ | 35 | |
| $R_e$ | 20 | |
| $R_i$ | 70 | |
| $L_p$ | 25 | |
| $L_n$ | 30 | |

Thus, the Jacobian matrix J was obtained by taking partial derivatives of the replicated dynamic equations, and then the parameters of the game model were assigned according to the criteria of the stable point of the evolutionary game: $\mathrm{tr}(J) < 0, \det(J) > 0$ and the actual situation of elderly-friendly renewal in Baoshan District, Shanghai.

Combined with the stability analysis of equilibrium point and the actual situation, the parameters of the game model were assigned and simulated.

The results of 50 evolutions can be seen from Figure 3. The simulation results can be obtained as follows. $E_{11}(0, a_2, a_3)$ is an unstable point. However, when conditions are met, it is an asymptotically stable point. Secondly, $E_1(0, 0, 0), E_5(1, 1, 0)$ are stable points. That is, (the government's loose management, enterprises' passive participation, residents' negative participation), (the government's active management, enterprises' active participation, residents' negative participation). $E_1$ means early stage of promotion; at this time, the government and enterprises are in a state of confusion, and residents are in a passive state. Therefore, three parties at this stage will adopt a passive strategy. At this point, it is necessary to rely on government departments to learn from excellent cases of government departments at the same level in community renewal to break the deadlock and create their own community renewal programs through continuous practice. $E_5$ represents the later stage of community renewal. Right now, the government and enterprises have a relatively complete system, but at the moment, residents' participation is low. The party organizations and grassroots community organizations should organize residents more congress, widely collect residents' opinions, establish resident autonomy organization, and improve residents' participation to ensure the sustainable development of community updates.

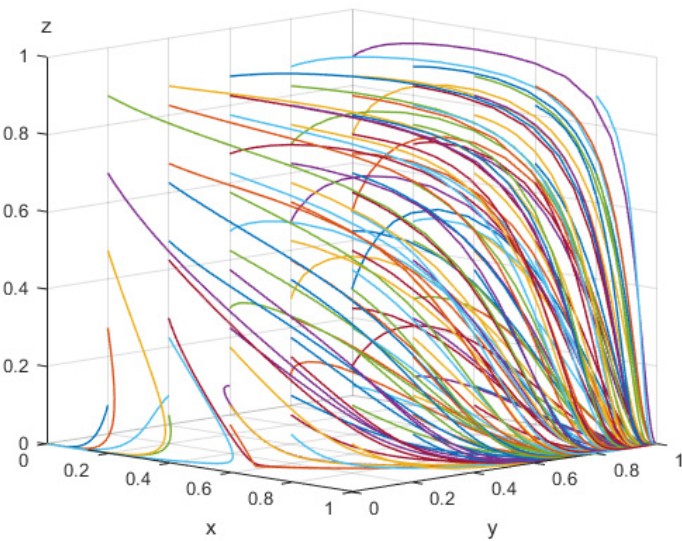

**Figure 3.** The result of 50 evolutions by initial values.

### 4.2. Stakeholders' Demands in Each Segment

In order to study the influence of the amount of subsidy ranted by the government during active governance on the other two parties, the initial value of S was changed from 20 to 25, 15, 10, and 5, respectively. As shown in Figure 4, after the promotion of subsidies, subsidies offset some of the cost and a reduction of benefit due to reducing the price of public service. So, the probability of enterprises' providing quality services ascends, and the community can optimize more residents based on the actual demand of the service and the subsequent operations in the process of transforming. After residents experience the significant improvement of the quality of life, their participation will increase greatly, accelerating the advancement of community renewal. However, when government subsidies are provided too much, the market will too strongly depend on the government's subsidies, which leads to the lack of the development of new technology. This makes residents not to obtain better services, so the participation of residents decreases. Besides, excessive fiscal causes the financial sectors hold opposite opinion and take negative strategy, which is not conducive to promoting community renewal. Therefore, government subsidies should not be excessive in order to promote market motivation, to develop new technologies, and to increase residents' participation and autonomy.

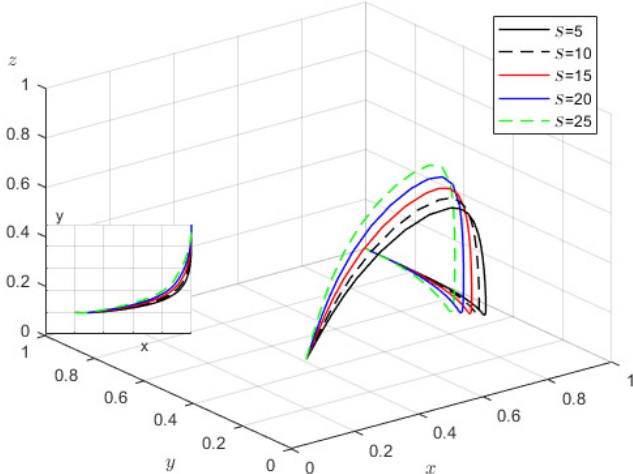

**Figure 4.** The impact of government's subsidy.

In a disguised way, the reduction of the number of subsidies increases the intensity of punishment, which also promotes the market sectors to be more inclined to participate

actively and converge faster. However, high penalties will cause enterprises to negatively participate, and low working efficiency makes community renewal efficiency drop so that residents are in a state of wait-and-see and even more choose not to participate. Therefore, the government needs to develop appropriate reward and punishment measures while promoting enterprises to participate actively, which can not only guarantee the subsidy spending plan to pass financial sectors' examination and approval, but also prevent the negative strategies of enterprises and make up part of government expenditure through punishment measures, thus forming a good situation of active government supervision and active participation of enterprises and residents.

By changing the value of $L_p$, we can observe the change of enterprises' strategy after the change of compensation intensity stipulated by the government. As shown in Figure 5, before the government positive management rate is one, community residents actively feedback enterprises' inaction through information feedback platform established by the government due to an unobvious improvement of their quality of life compared with an earlier one and even a decline of life satisfaction. Residents will require the government to intervene to urge enterprises to provide better services. After the government-positive management rate stabilizes at 1, residents' participation rate drops sharply because their needs are met. Enterprises continue to provide high-quality services for residents to avoid being punished by the government, so that enterprises' active participation rate stabilizes at 1. However, with the advancement of community renewal, residents are required to participate in the work more actively in later stage and to give real-time feedback on service usage experience. Therefore, community organizations can improve residents' participation by organizing regular discussion meetings and establishing user experience feedback platforms.

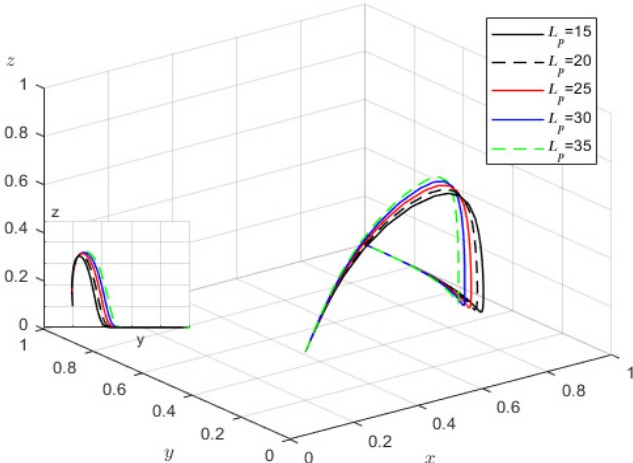

**Figure 5.** The impact of enterprises' compensation amount.

As a link between the government and residents, community planners play a role as a communication channel. With the help of community planners, the government can save part of the cost invested in community renewal (see Figure 6). Secondly, with the help of professional knowledge, community renewal planning is more reasonable and can bring residents a better life experience. Although the participation of community planners to some extent reduces the income of enterprises and brings some conflicts to enterprises, it means that the more community planners are involved, the slower the rate of active participation of enterprises reaches 1. However, under the regulation of the government, community planners can cooperate well with enterprises and give full play to their advantages to provide better services for residents. Therefore, the active participation rate of enterprises will eventually stabilize at 1. The participation of community planners increases the communication and feedback channels of residents in the process of renewal promotion, thus increasing residents' participation until their needs are finally met.

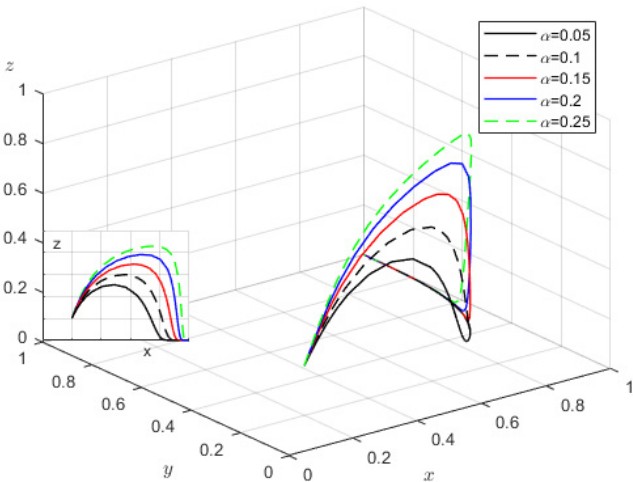

**Figure 6.** The impact of the community planner.

The performance rewards and social benefits from upper government departments to lower government departments, as well as the punishment from upper government departments to lower government departments and the loss of social reputation when the governance effect is poor, have an important impact on the government's strategy selection. It can be seen from Figure 7 that excessively high reward and excessively low punishment will cause the government's negative attitude in the process of promoting the renewal and loose management to enterprises, resulting in poor effect of community reconstruction. The improvement of residents' participation is mainly manifested in complaining to the upper government departments in order to make up for their losses. When rewards and punishments are equal, the government's rate of active governance will stabilize at 1 faster. Therefore, superior departments should reasonably allocate work to subordinate departments, formulate a performance evaluation system, regularly check the completion of tasks, and urge all departments to actively participate in the supervision and promotion of work. The active governance of the government will also promote the participation of enterprises so that the active participation rate of enterprises will stabilize at 1 faster, and enterprises will provide better services for residents.

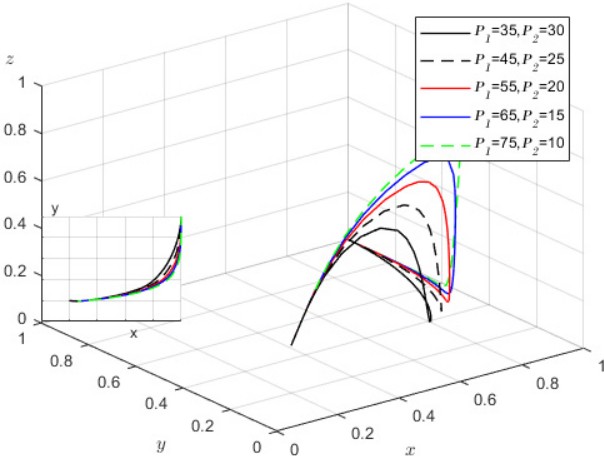

**Figure 7.** The impact of government rewards and punishments.

The cost paid by enterprises in community renewal is mainly composed of human and material resources, the price of resources and service which is mainly determined by the development and reform department. Therefore, we can adjust three parameters $C_1, C_2, R_f$ to observe the impact of market regulation on the three parties. (see Figure 8) The development and reform department are a government department, so changing these

three parameters will not have a great impact on the government's strategy selection, and the active governance rate of the government basically stays stable at 1 at the same speed when parameters differ. After development and reform department cut the price of the resources needed by community renewal and control the enterprises' services price setting, enterprises will be more actively involved in community renewal due to the cost reduction. At the same time, residents' willingness to purchase services will rise due to lower service prices. So, the government can regulate the market to achieve the purpose of promoting enterprises and residents to actively participate in community renewal.

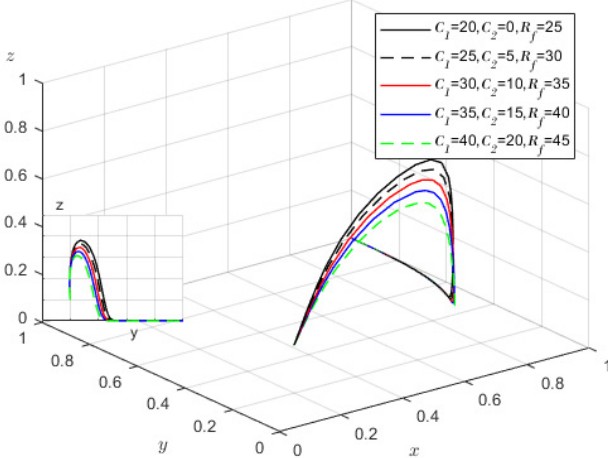

**Figure 8.** The impact of market regulation.

The government's initial strategy was adjusted to 0.2, 0.5, and 0.7, (Figure 9) respectively, and the initial strategy values of enterprises and residents were adjusted on this basis. It can be found that the more the government prefers to adopt active governance strategy at the initial stage of community renewal, the faster the strategy selection will be stabilized at 1. Moreover, the more enterprises and residents cooperate with the government to promote the work, the easier it is for the government to adopt relevant resolutions and achieve active cooperation among various departments, which is conducive to the formulation of community renewal plans and the promotion of work.

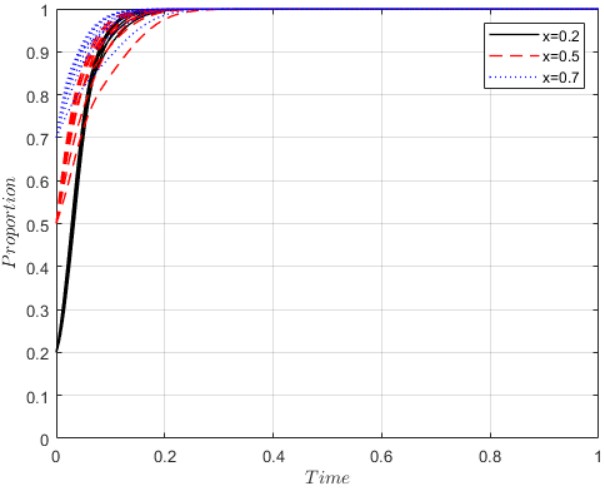

**Figure 9.** Evolutionary diagram when x differs.

The initial strategy values of enterprises were respectively adjusted to 0.2, 0.5, and 0.7. It can be seen from Figure 10 that the active participation rate of enterprises stabilizes at 1 at a faster pace, in which the leading model of enterprises plays a large role. When enterprises cooperating with the government obtain subsidies and other outsourcing

projects except community renewal, the remaining unparticipating enterprises will join the work one after another because of the bellwether effect so that the government can select the enterprises that can provide relatively better service and cooperate with them. When residents' participation increases, good communication between enterprises and residents' representatives enable enterprises to transform communities according to residents' needs in the process of community reconstruction, faithfully meeting residents' living needs and improving their quality of life.

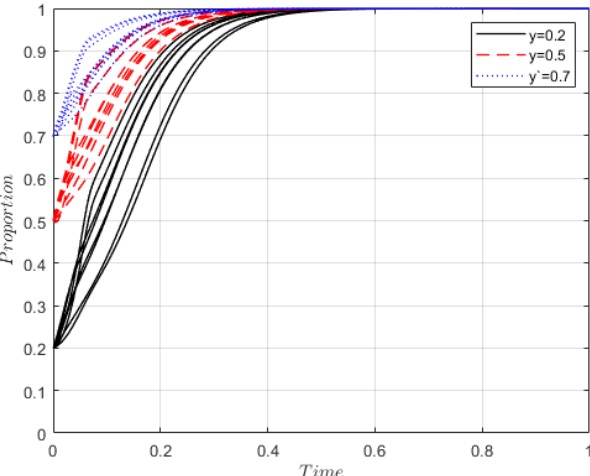

**Figure 10.** Evolutionary diagram when y differs.

The initial strategy values of residents were adjusted to 0.2, 0.5, and 0.7, respectively. It can be seen from Figure 11 that the higher the initial participation of residents, the slower the subsequent participation rate will stabilize at 1. The higher the probability that the government takes active governance and the enterprises take active participation, the slower the decline of residents' participation is. Community renewal is closely related to residents, so residents' participation should be gradually improved. But the lack of experience feedback platform, week personal influence cause residents' participation to decline continually. They can only passively accept the government's management and enterprises' service provisions; residents are unable to fully express their appeals to really improve their life quality. Therefore, the government can increase residents' participation by introducing community planners, establishing an information feedback platform, and holding lectures and opinion consultation meetings to ensure the sustainable development of community renewal at a time.

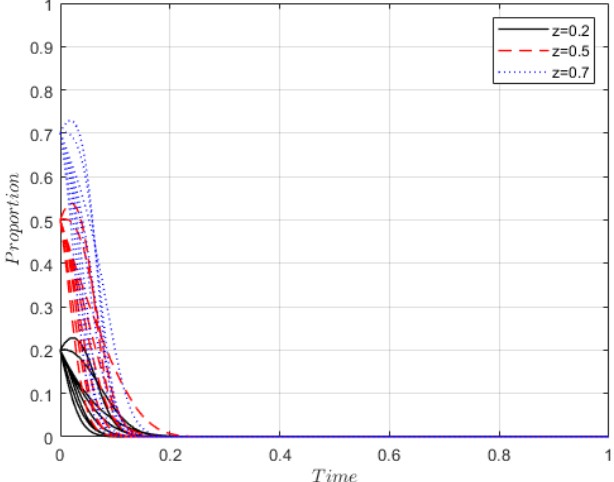

**Figure 11.** Evolutionary diagram when z differs.

*4.3. Stakeholder Characteristics and Cooperation Methods*

The community planner system in China is characterized by community roots, professional intervention, and balanced demands, and it plays a role in making decisions and advising on the renewal of ageing in the community, reconciling the demands of various stakeholders. Based on China's practical experience, it can be argued that government departments, enterprises (market social organizations), and residents are the key stakeholders whose attitudes and chosen strategies greatly impact age-friendly regeneration.

The government wants to achieve regional economic benefits and promote effective community governance in age-friendly regeneration. Therefore, they provide support policies, subsidies, and other facilities to motivate social organizations and residents to actively participate, and provide employment opportunities to improve and enhance the image of grassroots organizations, thus realizing economic and social benefits.

## 5. Discussion

In general, this study aimed to improve the efficiency of urban planning and management decision-makers, and provide a more effective way for the public to participate in community construction and management. The improvement of workflow in both urban management and public participation helps to deal with the contradiction in local stakeholders, including government, communities, and investors. According to Yang et al. (2015), all the stakeholders play critical roles in acting as key players and decision-makers in the rapid development of the urban environment [26]. Moreover, to achieve a better social, cultural, economic, and environmental role for urban spaces, the link between urban management and effective place governance practices becomes increasingly important [25,28]. In order to achieve a balanced structure between different stakeholders, the research also promotes community resource sharing and environmental co-creation. The sustainable development of both the urban environment and Cyberspace cannot be achieved without the support of local residents, and the residents of places with quality local resources cannot do without the involvement of local communities [26,29]. Therefore, it is critical to increase residents' attention to and participation in community public affairs to form corporative engagement in the development. On the other hand, the research showed that the incorporation of multiple stakeholders, including local governance and other decision-makers, could contribute to forming comprehensive conceptual frameworks for further analysis in Cyberspace, where the social issues could be addressed with more preventative approaches [9]. Therefore, the research realizes a smart transformation of community renewal.

*5.1. Digital Transformation Requires Balancing the Interests of Multiple Parties*

There are differences between enterprises, residents, and the government at the level of demand. Among those three stakeholder groups, enterprises are capital-driven and aim to use privatization to capture the potential value of community space and want to obtain the maximum benefit in exchange [22,23,30]; the government wants to maintain a fair and equitable distribution of community resources and eliminate potential externalities in the community through a stable control of spatial resources [30,31]; and resident groups want to negotiate among themselves to obtain the maximum value from the use of community resources [32,33].

Therefore, the digital transformation needs to achieve a balance of interests between multiple parties in order to facilitate the smooth implementation of community regeneration and to generate sustainable regeneration benefits. In order to achieve a balanced interests among multiple stakeholders, Emery and Trist (1965) and Trist (1983) argue that organizations must move from intra-organizational goal focus to identifying goals and goal paths that maximize the interests of all parties within the organization [26,30,34]. According to Pedol et al. (2021), a structure with balanced interests and sustainable values shared by each stakeholder group contribute to actively working to meet the needs of individuals and promote best practices in sustainable development.

## 5.2. Digital Transformation Requires Adequate Participatory Coordination Mechanisms

The simulation process and the results of the evolutionary game fully confirm the significance of the multi-party participation and coordination mechanism, which can facilitate effective communication between multiple parties and reach a balance of interests as soon as possible [33,34]. The digital transformation of communities will accelerate the reorganization of power structures and the widespread promotion of public participation in communities [26,35]. The new resources represented by information technology have given multiple parties channels to participate fully, forming a typical decentralized model, which makes it all the more necessary to establish a coordination mechanism for participation as soon as possible [26,36].

Government is the central subject of environmental governance, playing the role of helmsman, servant, and regulator. Enterprises and public organizations are important participants in marine environmental governance. Businesses enterprises and public organizations are important players in the governance of the marine environment. Government coordinates with and guides business and public organizations from seeking their own goals to contributing to the overall direction of direction of government [37]. Marine enterprises, as important subjects of governance, are active participants in improving the marine environment. As the main cause of pollution in the marine environment, marine enterprises are an important target for government intervention and an important source of support for marine environmental protection [30]. The relationship between government and public organizations has changed from one of disassociation to one of cooperation to maximize the delivery of public services relating to the marine environment. The involvement of public organizations is a dominant trend in social governance. As public organizations are direct victims of environmental externalities, they assume the role of participants and watchdogs. They not only have strong internal power to change environmental conditions, but also seek to protect the marine environment by actively participating in marine environmental protection and sharing responsibility with governments and marine enterprises [38,39].

In the process of digital transformation, more digital tools can be introduced to structurally change the position of the government, residents, and enterprises in the communication scenario, to facilitate extensive tripartite participation in all aspects of community renewal, and to empower stakeholders to make decisions [40,41]. For example, mobile internet-based voting and proposal tools, VR-based spatial experience tools, and online community operation tools can be used to form a community of tripartite participation that connects online and offline to effectively organize issues, book discussions, and share opinions.

## 5.3. Digital Transformation Requires Intelligent Governance and Decision Aids

In the process of digital transformation of communities, community stakeholders are characterized by close ties and complex demands, which strengthen the perception of individual citizens of the quality of governance, and small mistakes in decision-making may be combined with other risks, causing difficulties in community renewal and governance [42,43]. As a result, the government is faced with more complex and multifaceted decision-making challenges that require it to be more scientific, dynamic, and forward-looking. This requires an expanding capacity to collect, process, and make decisions on community development demands and other relevant information [44,45].

Based on the elaboration of the multiparty interest balance and participation and coordination mechanism in the aforementioned chapters, it can be considered that evolutionary game modeling and simulation is a key technology for intelligent governance and decision support, which can be compounded with technologies such as big data, cloud computing, and artificial intelligence to form an accurate community governance model [43,46]. On this basis, real-time data can be accessed to realize the risk perception and monitoring of the community's full claim elements, the whole governance process and the whole development scenario, realizing the digital twin, presenting an accurate and comprehensive picture of the problems faced by community renewal and governance and revealing universal laws [47,48]. This can facilitate the transformation of government

decision-making and governance from passive response to active foresight and avoidance, empowering community development.

## 6. Conclusions

The initial core model of the evolutionary game function module was developed based on the simulation design, and a pathway of strategies output and decision support was developed that can be applied in the future of community governance in the digital twin. Furthermore, the formation of a growing database of autonomous social attribute demand expression analyses and strategy optimization evolutionary game simulations of the auxiliary decision-making function module dovetails with the strategy support platform for city operation, urban management, and community management. The main significance of this paper is that (1) the framework approach, proposing an autonomous decision-making system based on reinforcement learning that provides optimal strategies for the multidimensional intersection of sociology and traditional spatial planning and government governance, proposes object-based research and development and scenario deepening, innovative ideas in urban refinement and governance, and has important implications for solving self-optimization problems in related urban operations and urban management in the future. (2) Due to the systematic framework requirements, the whole is divided into four types of communities, and different communities present different benefits in different renewal scenarios. Moreover, this paper set parameters and simulations for one of the communities, and validated the game behavior and results, which has theoretical and practical significance for urban planning and lays the foundation for access to the overall framework. (3) The importance of social and perceptual dimensions in the digital twin in the cyber era was proposed. Based on computational social science and evolutionary game theory, we examined the evolution of social governance state and strain decision models, built a simulation method for the evolution of complex systems of social governance driven by the fusion of data and knowledge, proposed a system response to residents' ubiquitous perception and ubiquitous participation in the cyber era, and finally, enhanced the internal drive for comprehensive and sustainable urban development.

**Author Contributions:** Conceptualization, Y.Z.; methodology, X.Z. and C.L.; software, J.Z.; validation, X.Z., Y.X. and S.W.; formal analysis, Y.Z; investigation, H.L.; resources, Y.X.; data curation, Y.Z.; writing—original draft preparation, Y.Z., H.L. and X.Z.; writing—review and editing, Y.Z. and H.L; visualization, S.W.; supervision, C.L. and Y.Z.; project administration, Y.Z.; funding acquisition, Y.Z. and Y.X. All authors have read and agreed to the published version of the manuscript.

**Funding:** This research was funded by the Ministry of Housing and Urban-Rural Development of the People's Republic of China (2021-K-148), and the National Natural Science Foundation of China (52208079).

**Conflicts of Interest:** The authors declare no conflict of interest.

## Appendix A

A1. The solution of equilibrium points of the three-way game system

$$E_1 = (0,0,0); E_2 = (1,0,0); E_3 = (0,1,0); E_4 = (0,0,1); E_5 = (1,1,0);$$

$$E_6 = (1,0,1); E_7 = (0,1,1); E_8 = (1,1,1); E_9 = \left(0, \frac{R_f}{L_n+R_i}, \frac{C_1-C_2}{M+R_e}\right);$$

$$E_{10} = \left(\frac{L_n+R_i-R_f}{L_n+(2-\alpha)R_i}, 1, \frac{S-P_1-\beta C_e}{P_2+\alpha C_e}\right); E_{11} = \left(\frac{R_f}{L_p+\alpha R_i}, 0, \frac{S-M}{P_1+P_2+\alpha C_e}\right);$$

$$E_{12} = \left(\frac{C_1-C_2}{M+R_e}, \frac{S-M}{P_1-M+\beta C_e}, 0\right); E_{13} = \left(\frac{C_1-C_2-M-R_e}{L_p}, \frac{M+P_1+P_2-S+\alpha C_e}{M-\beta C_e}, 1\right)$$

A2. The Jacobian matrix of this study:

$$
J = \begin{bmatrix} J_1 & J_2 & J_3 \\ J_4 & J_5 & J_6 \\ J_7 & J_8 & J_9 \end{bmatrix} = \begin{bmatrix} \frac{\partial F(x)}{\partial x} & \frac{\partial F(x)}{\partial y} & \frac{\partial F(x)}{\partial z} \\ \frac{\partial F(y)}{\partial x} & \frac{\partial F(y)}{\partial y} & \frac{\partial F(y)}{\partial z} \\ \frac{\partial F(z)}{\partial x} & \frac{\partial F(z)}{\partial y} & \frac{\partial F(z)}{\partial z} \end{bmatrix}
$$

$$
= \begin{bmatrix} (1-2x)\left[\begin{array}{c}(P_1 - M - \beta C_e - P_1 z)y \\ +(P_1 + P_2 + \alpha C_e)z + M - S\end{array}\right] & x(1-x)(P_1 - M - \beta C_e - P_1 z) & x(1-x)[(-P_1 y) + P_1 + P_2 + \alpha C_e] \\ y(1-y)\left[M + R_e - (M - L_p + R_e)z\right] & (1-2y)\left[\begin{array}{c}(M + R_e - (M - L_p + R_e)x)z \\ +(M+R_e)x + C_2 - C_1\end{array}\right] & y(1-y)\left[(L_p - M - R_e)x\right] \\ z(1-z)\left[-(L_n + L_p + 2R_i)y\right] & z(1-z)\left[L_n + R_i - (L_n + L_p + 2R_i)x\right] & (1-2z)\left[\begin{array}{c}(L_n + \alpha R_i - (L_n + L_p + 2R_i)y)x \\ +(L_n + R_i)y - R_f\end{array}\right] \end{bmatrix}
$$

A3. The relationship of $a_1 a_2 a_3$ is shown as below:

$$
\begin{aligned}
a_1 = & -(M^{\wedge}2 * Rf - Ln * M^{\wedge}2 - M^2 * Ri - C1 * Ln * P1 - C1 * Ln * P2 + C2 * Ln * P1 + \\
& C2 * Ln * P2 + C1 * P1 * Rf - C2 * P1 * Rf - C1 * P1 * Ri - C1 * P2 * Ri + C2 * P1 * Ri + \\
& C2 * P2 * Ri - Ln * M * Re + Ln * M * S - M * P1 * Rf + M * Re * Rf - M * Re * Ri + Ln * \\
& Re * S + M * Ri * S - P1 * Re * Rf + Re * Ri * S - C1 * Ce * Ln * alpha + C2 * Ce * Ln * \\
& alpha - C1 * Ce * Ri * alpha + C2 * Ce * Ri * alpha + Ce * M * Rf * beta + Ce * Re * Rf * \\
& beta) / (Re * Ri + Ln * M + Ln * Re + M * Ri)
\end{aligned}
$$

$$
\begin{aligned}
a_2 = & (M^{\wedge}2 * Rf - Lp * M \wedge 2 - C1 * Lp * P1 - C1 * Lp * P2 + C2 * Lp * P1 + C2 * Lp * P2 - \\
& Lp * M * Re - Lp * M * Rf + Lp * M * S + M * P1 * Rf + M * P2 * Rf + M * Re * Rf + \\
& Lp * Re * S + Lp * Rf * S - M * Rf * S + P1 * Re * Rf + P2 * Re * Rf - Re * Rf * S - \\
& M^{\wedge}2 * Ri * alpha - C1 * Ce * Lp * alpha + C2 * Ce * Lp * alpha + Ce * M * Rf * alpha - \\
& C1 * P1 * Ri * alpha - C1 * P2 * Ri * alpha + C2 * P1 * Ri * alpha + C2 * P2 * Ri * alpha + \\
& Ce * Re * Rf * alpha - M * Re * Ri * alpha + M * Ri * S * alpha + Re * Ri * S * alpha - \\
& C1 * Ce * Ri * alpha^{\wedge}2 + C2 * Ce * Ri * alpha^{\wedge}2) / (Lp * P1 + Lp * P2 + Ce * Lp * alpha + \\
& P1 * Ri * alpha + P2 * Ri * alpha + Ce * Ri * alpha^{\wedge}2)
\end{aligned}
$$

$$
\begin{aligned}
a_3 = & (2 * Ln * M * Rf - 2 * Lp * M * Rf + Ln * P1 * Rf + Ln * P2 * Rf - Lp * P1 * Rf - \\
& Lp * P2 * Rf - 2 * Ln * Rf * S + 2 * Lp * Rf * S + Ce * Ln * Rf * alpha - Ce * Lp * Rf * \\
& alpha) / (Lp * P1 + Lp * P2 + Ce * Lp * alpha + P1 * Ri * alpha + P2 * Ri * alpha + Ce * \\
& Ri * alpha^{\wedge}2)
\end{aligned}
$$

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
