# Peer review of "Using the Dual Concept of Evolutionary Game and Reinforcement Learning in Support of Decision-Making Process of Community Regeneration—Case Study in Shanghai"

_buildings, doi:10.3390/buildings13010175_

Round 1

Reviewer 1 Report

The article takes up a very important aspect of the life of the inhabitants of a given urban space in the aspect of dynamically developing digital technologies. The basis of the adopted scientific narrative is digital simulation technology, which is increasingly used in the field of modern city management. Its purpose is to provide multi-scenario simulation and decision support, because there are multidimensional optimization conflicts of stakeholder groups. It is good that the authors clearly appreciate the beneficial transformation of a separate area of the commune (in economic, social, architectural and planning aspects), which is in a state of crisis resulting from the occurrence of economic and social factors. The local government's responsible and committed approach to building a strong and sustainable economy is based on the idea of a digital twin city.

The proposed Title, unfortunately, is not scientific, but rather journalistic. The form of the question (consistent with the possible research question) should be changed into the form appropriate for a scientific article. For example - "Using the dual concept of Evolutionary Game and Reinforcement Learning in support of Decision-Making Process of Community Regeneration - case study in Shanghai".

The Keywords used do not clearly emphasize the digital aspect, so it should be better exposed by adding, for example, "digital technology", "digital twin city", "AI technology-driven community regeneration". In Abstract, the research problem was clearly indicated, but the proposed way of solving it seems to be too large and multi-layered in its description. The limitations of existing research should be pointed out, and authors must explore the originality and contribution of the research in this section.

In the Introduction, the authors correctly introduce the problem of changes in the perception of the contemporary social formation as a result of the dynamic development of computer science and information technologies. This is a part about good logic of arguments, which in addition to justifying the research problem, emphasizes the stages of the research process, i.e.: analysis of the evolution of decisions - proposal of an evolutionary simulation approach - data extrapolation - the use of Big Data in creating the architecture of the digital twin system. However, the research problem must be better (clearly) defined, and the purpose and contribution of the work should be outlined better.

Part entitled "2. Literature review” is correct, but it should be improved. It should be devoted to presenting a critical analysis of the latest works related to this topic, justifying the purpose of the research. In addition, critical remarks should be made regarding the results of the cited works (although a significant number - 49 items - were cited in the bibliography). In 2.1. the terms "game theory" and digital twin should not be confused, especially given the further explanations in this section of the article. In 2.2. good arguments justifying the usefulness of "evolutionary game theory" were presented. Part 2.3. it is a theoretical preparation (ordered characteristics of different types of communities) for the "3.Methods" section. The hypothesis formulated in it is well posed: "The community planner system aims to achieve a balance of multiple interests in the practice of age-friendly renewal." But, for example, in section 2.3. the purpose of the article appears in a different version: „/…/This paper analyzes and sorts out the community types and benefits, sorting out the analysis and planning at the framework level, and simulates the analysis for the age-friendly renewal scenarios under the responsibility of community planners/…/”.

Part entitled "4. Evolutionary Game Analysis” contains the correct algorithm to follow to simulate operations in the context of community age-ap-propriate renewal practice in Baoshan District (Shanghai). The proposed comparative model is based on study the game process, strategy selection, and game results. The proposed scientific construct proves the usefulness of the proposed model for decision makers through governance strategies through analysis of stakeholder benefit relationships.

Also a successful part is "5. Discussion”, although one might expect that the authors would not formulate point 1 in the following wording: "Balancing the interests of multiple parties in digital transformation", because the need is a research assumption and not a result (especially since foreign opinions are invoked). The next point "Digital transformation requires adequate participatory coordination mechanisms" and "Digital transformation requires intelligent governance and decision aids" are better documented because they answer the question: "what is required?".

In the last part of "6. Conclusion”, the authors clearly indicated their undeniable contribution to science, systematizing it in three key points.

Summing up the review, it should be emphasized that the article is well written from both a formal and methodological point of view. It carries a significant cognitive value and encourages further scientific discussion. The adopted narrative and scientific process is also appropriate and well documented. The comments raised are only intended to improve its formal and methodological dimensions.

Reviewer 2 Report

Comments

In this paper, the authors have tried to explore the social attributes and interactions in the context of digital twin related research and development for city level planning, management and overall decision making. For that purpose deep reinforcement learning and evolutionary game modelling is utilized. The concept is unique and interest to the readers and could be recommended for publication. However the reviewer has following concerns.

Following please find comments prior to possible publication.

1.      Please highlight the novelty of your work as much of the research has been conducted in this area.

2.      The paper is more related to the digital twin but the title of the paper does not reflect that. Please clarify?

3.      Also, how digital twin is defined is confusing. Digital twin require bidirectional data and decision making. Please provide relevant reference in answer to the comment regarding definition of digital twin in the context of city level decision making?

4.      How precisely the reinforcement learning is incorporated into the evolutionary game theory. Little for information is required.

5.      The numbering of references is kind of odd in the manuscript. The higher order references are coming before. The order of numbering should be like 1,2,3,4 . .. and not 30,31,3,4,6.
